# Inter-Relationships between Test Weight, Thousand Kernel Weight, Kernel Size Distribution and Their Effects on Durum Wheat Milling, Semolina Composition and Pasta Processing Quality

**DOI:** 10.3390/foods9091308

**Published:** 2020-09-16

**Authors:** Kun Wang, Bin Xiao Fu

**Affiliations:** Grain Research Laboratory, Canadian Grain Commission, Winnipeg, MB R3C 3G7, Canada; kun.wang@grainscanada.gc.ca

**Keywords:** durum wheat, milling quality, kernel size, kernel weight, test weight, semolina, pasta color, pasta texture

## Abstract

Although most of the durum wheat produced in the Canadian prairies in 2017 and 2018 met the test weight (TW) requirements for the top grades of Canada Western Amber Durum (CWAD), some samples of top grades were inferior in milling quality. To understand the abnormality, this study was conducted to investigate TW, thousand kernel weight (TKW) and kernel size distribution (KSD) in relation to durum milling potential, semolina composition and pasta quality. With reduction of kernel size, semolina and total milling yields decreased progressively, and kernels passing through no.6 slotted sieve had detrimental impact on milling. The overall relationship between TW and milling yields appeared to be genotype dependent. At similar TW, variety showed lower milling yields had greater proportion of smaller kernels. By account for the difference in KSD, greater relationships (*R*^2^ > 0.91, *p* < 0.001) were found for TKW and proportion of kernels passing No.6 slotted sieve with milling yields than TW (*R*^2^ = 0.75, *p* < 0.001). This infers potential use of small kernels (passing No.6 slotted sieve) as a new objective grading factor for rapid prediction of milling quality of CWAD. Although small kernels exhibited much higher yellow pigment than the larger ones, pasta made from small kernels was duller, redder and less yellow, likely due to the higher semolina ash and protein contents, which adversely affected pasta color.

## 1. Introduction

Test weight (TW) is widely used as a primary specification in wheat trading and generally accepted by milling industry as an indicator of milling potential. However, the relationship between TW and wheat milling potential is not always warranted and generally affected by wheat classes, varieties within the class and specific growing condition [1,2,3]. Studies have shown that TW was affected by wheat moisture, kernel density, kernel shape and packing factors, which were not related to milling yield [4,5,6,7,8,9,10].

Significant research has been conducted to understand the relationship between wheat physical characteristics and milling quality of common and durum wheat [2,3,4,6,7,8,9,10,11,12,13,14,15,16,17]. Hook (1984) found that the correlation between TW and flour yield was poor and TW could not be used for predicting flour yield for UK winter and spring wheats [4]. The relationships were affected by growing year, growing site, variety and endosperm texture. Lyford et al. (2005) successfully used parameters (i.e., kernel weight and kernel hardness) generated with single kernel characterization system and TW to predict flour extraction rate of US hard red winter wheat (*R*^2^ = 0.81) across a wide range of growing environment [13]. By evaluating 92 Canadian durum harvest composite samples from 1984 and 1985, Dexter et al. (1987) found TW was highly positively related to thousand kernel weight (TKW) (*R*^2^ = 0.88) [3]. Moderately strong to strong relationships (*R*^2^ = 0.47–0.77) were shown between TW to semolina yield and milling score. The authors concluded that TKW alone or in combination with TW did not offer greater advantage over TW in predicting durum wheat quality likely due to the strong interdependence of TW and TKW. Dexter and Symons (2007) emphasized that there were strong relationships between TW and TKW to semolina and granular yields [11]. However, the relationship was strongly biased by growing year at a given level of TW or TKW, therefore weakening the relationships when data from all three years were pooled. It appears the inter-relationships between TW and TKW and their associations with milling yields were not conclusive. Conflicting results have been reported when predicting durum and common wheat milling quality with different kernel physical parameters.

In Canada, TW greater than 80.0 and 79.0 kg/hL is required to qualify for No.1 and No.2 grades (export) of Canada Western Amber Durum (CWAD) wheat class, respectively [18]. Millers expect CWAD with high TW to possess superior milling quality and produce high yield of granular semolina with minimum production of fine flour [19]. Due to the overall dry growing conditions in 2017 and 2018, majority of durum wheat produced in the Canadian prairies was high in protein content and met the TW requirements for the top grades of CWAD. However, some samples of the top grades (>No.2 CWAD) exhibited smaller kernel characteristics and were inferior in milling performance. To protect the milling quality of top grades of CWAD, a thorough investigation is required to justify the use of TW and potential other alternative quality parameter in differentiating milling quality of amber durum wheat.

The objective of this study was, therefore, to investigate the inter-relationships between TW, TKW and KSD and their effects on durum wheat quality with emphasis on durum milling potential. The ultimate goal of this study was to identify an objective grading factor that can better reflect the milling quality of durum wheat than TW.

## 2. Materials and Methods

### 2.1. Wheat Samples

Two sets of durum wheat samples were used in this study. Set I was based on a composite of 14 commercial cargo loading samples, representing over a quarter million metric tons of durum wheat in a shipping period of eight months. To investigate the relationship between kernel size and functionality, the composite was fractionated into five different kernel sizes using a Carter dockage tester (Simon-Day Ltd., Winnipeg, MB, Canada) equipped with a series of slotted sieves (no.8, 7, 6 and 5 with apertures of 3.18, 2.78, 2.38, and 1.98 mm × 19.05 mm, respectively). Set II consisted of 21 composites of four major CWAD varieties (Transcend, Brigade, Strongfield, and CDC Verona) prepared from harvest samples submitted by producers from 2017 growing year. For each variety, samples were grouped and composited based on their TW and protein content. Each composite was prepared from a minimum of 15 samples. Detailed descriptions of sample sets I and II were presented in Table 1 and Table 2, respectively.

### 2.2. Wheat Physical Properties

Weight per hectoliter, or TW, was measured with a Schopper Chondrometer with a 1L container. TKW was measured with an electronic seed counter (Model 750, The Old Mill Company, Savage, MD, USA) using 20 g of samples, from which all broken kernels were manually removed. KSD was determined with a series of slotted sieves (no.5, 6, 7 and 8). One hundred grams of wheat was subsampled and manually shaken for 30 s, after which the five fractions separated by the sieves were weighted individually. All tests for wheat physical properties were conducted at least in duplicate.

### 2.3. Wheat Milling

All durum samples were milled into semolina in duplicate of 1.5 kg lots separately using a four stand Allis-Chalmers laboratory mill (West Allis, WI, USA) in conjunction with a laboratory purifier following the mill flow previously described by Dexter et al. (1990) [20]. The mill room was controlled at 21 °C and 60% relative humidity. Semolina is defined as having less than 3% pass through a 149-micrometre sieve. Total milling yield is the combination of semolina and flour. Both total and semolina yields are reported as a percentage of the cleaned wheat on a constant moisture basis. Semolina granules were prepared by adding the most refined flour stream(s) to semolina until 70% extraction was reached for analysis and pasta processing. To calculate semolina yield on constant ash basis, cumulative ash curve derived by Dexter et al. (2007) [11], who followed the same milling protocol was adopted. The expected semolina yield at constant ash content of 0.71% could be referred as: granules ash score (%) = semolina yield − ((granule ash − 0.71)/0.0116).

### 2.4. Wheat and Semolina Analysis

Protein content of whole wheat and semolina were measured following the method previously described by Williams et al. (1998) with a LECO Truspec N CNA (combustion nitrogen analysis) analyzer (Saint Joseph, MI, USA) [21]. Grounded wheat meal was prepared using Retsch ZM 200 mill (Retsch GmbH, Haan, Germany) equipped with a 0.5 mm screen (Trapezoid holes) at speed of 14,000 rpm. Ash content, wet gluten and gluten index were determined using AACC International approved methods 76–31.01 and 38–12.02, respectively [22]. Semolina color was measured with a Minolta colorimeter CR-410 (Konica Minolta Sensing, Inc., Tokyo, Japan) with a D65 illuminant. Color readings are expressed on the CIELAB color space system with L*, a* and b* parameters representing brightness, redness and yellowness values, respectively. A micro scale rapid extraction procedure as described by Fu et al. (2013) was used for determination of total yellow pigment (TYP) content of semolina [23]. Granulation of semolina was determined by a series of sieves in U.S. standard (425, 250, 180 and 150 µm). Semolina particle size distribution (PSD) was calculated based on the weight of each size fraction as a percentage of the total weight.

### 2.5. Spaghetti Processing and Measurement of Quality

Spaghetti was produced from semolina following the method of Fu et al. (2017) [24]. Semolina was mixed with water in a high speed asymmetric centrifugal mixer at water absorption of 31 to 32% to maintain constant extrusion pressure of about 100 bar. Dough crumbs were extruded through a four-hole Teflon coated spaghetti die (1.8 mm) with application of vacuum. The fresh pasta was subsequently dried in a pilot pasta dryer (Bühler, Uzwil, Switzerland) with a 325 min drying cycle and maximum temperature of 85 °C. To measure spaghetti color, a 6.5 cm band of spaghetti strands were mounted on a white mat board, and color was determined using a Minolta colorimeter as described above. 

Cooked spaghetti firmness was determined using the Stable Micro Systems TA.XT2i Texture Analyser (Texture Technologies Corp., Scarsdale, New York, NY, USA). Cooking time was fixed at 8 min with twelve spaghetti strands (5 cm in length) being cooked each time. Cooked spaghetti strands were drained and immediately aligned on the base plate. Five strands with no spacing were cut perpendicular at a fixed compression depth of 4.9 mm at a crosshead speed of 1 mm/sec with a TA-47 blade of diameter of 0.5 mm. Average peak force of six cuttings was reported.

### 2.6. Statistical Analysis

Statistical analysis was performed using the SAS v. 9.4 Software (SAS Institute Inc., Gary, NC, USA). Analysis of variance (ANOVA) was used to evaluate the impact of kernel size and test weight on durum wheat and semolina quality characteristics. For sample set I and II, each sample were treated as an independent sample. Tukey’s test following the analysis of variance indicated significant differences with a level of *p* < 0.05.

## 3. Results and Discussion

### 3.1. Impact of Kernel Size on Durum Wheat Milling Quality

To investigate the impact of kernel size on durum wheat physical and milling characteristics, a composite of cargo loading samples, graded as No.1 CWAD, was fractionated into five portions based on kernel size. Table 1 shows the wheat physical and milling properties of each segregation and the original unsorted sample. Significant impact of kernel size was found on wheat physical and milling properties with greater influence shown on TKW, granule ash score, TW, wheat protein content, and percentage of flour produced during milling as indicated by F values. With the decrease of kernel size, wheat protein content increased significantly from 14.1 to 18.6% accompanied by gradual decreases of TW from 82.6 to 73.5 kg/hL. The reduction in TW infers that the smaller kernels were significantly less dense than the corresponding larger ones although they might pack more compactly. Similar observation has been reported from previous research [3,11]. Compared with TW, greater influence of kernel size was found for TKW as shown by the greater F value (12,482.5 vs. 4797.1). TKW decreased more than three folds from the largest kernel fraction (61.9 g) to the smallest one (19.3 g), while the corresponding TW reduced by 9.1 kg/hL.

Table 1 shows that semolina (1.2–1.5%) and total milling yields (1.1–2.3%) reduced gradually from fraction retained by no.8 sieve to that of above no.7 and no.6 sieves. Further decrease in kernel size (passing through no.6 and no.5 sieves) resulted in a drastic decrease in semolina (2.7%) and total milling yields (2.7 to 2.9%) accompanied by sharp increases in wheat (0.12–0.25%) and semolina ash content (0.09–0.15%). It appears that the critical kernel size at which durum milling quality was severely affected was under 2.38 mm kernel diameter (aperture of no.6 slotted sieve) or TKW below 36 g in the current research. Dexter et al. (2007) observed comparable adverse effects of small size kernels on durum milling quality with a threshold of about 40 g in TKW, below which milling quality deteriorated drastically [11]. The higher TKW threshold reported by Dexter et al. (2007) was likely due to the much lower wheat protein content (by 2–5%) of their samples as compared to those used in this study.

To further understand the impact of test weight on durum milling quality, the relationships between TW and semolina yield and granule ash score (constant ash basis) for sample sets I and II were illustrated in Figure 1a,b, respectively. From Figure 1a, TW was highly positively associated with semolina yield and granule ash score for selected wheat kernel fractions. Low TW resulted in reduced yield and higher ash content in semolina. When semolina yield was calculated on a constant ash basis, the higher ash content led to about 3.5% decrease in semolina yield with each one kg/hL decrease in TW, significantly higher than 0.8% without adjusting for ash content. Similar trend was shown for sample set II (Figure 1b) despite the relationships between semolina yields and TW were weaker. Furthermore, analysis of semolina PSD (Table 1) showed that samples with lower TW produced significantly higher amount of flour (<150 µm) and less coarse semolina (>250 µm) than the corresponding larger kernels. Kernel size has little impact on the proportion of fine semolina (150–250 µm) except that kernels passing through no.5 slotted sieve had significantly higher amount of fine semolina.

### 3.2. Effect of Genotype on the Relationship between TW, TKW and KSD

Figure 2 presents the effect of genotype on the relationship between TW and TKW for the varietal composites in sample set II. Although TW was highly related to TKW (*r* = 0.92, *p* < 0.001), the relationship was affected by the durum varieties selected. At a given level of TW, variety Transcend exhibited significantly lower TKW than other three selected varieties across a wide range of TW evaluated (*p* > 0.05). In other words, Transcend showed consistently higher TW at a constant TKW, suggesting factor(s) other than kernel weight and/or size contribute to the relatively higher TW of Transcend.

The physical and milling properties of the four varietal composites were summarized in Table 2. With the decrease of TW and TKW, the reduction in kernel size was evidenced by the drastic drop in the proportion of kernels retained above no.8 and no.7 slotted sieves, accompanied by large increases in the percentage of kernel fractions passing through no.6 and no.5 sieves. By analyzing KSD of the varietal composites (Table 2), at equivalent TW, Transcend possessed significantly greater proportion of small kernels (through no.6 slotted sieve) and less large ones (above no.7 sieve) than the others, especially at low TW range (Figure 3). Consequently, this led to lower TKW for Transcend as compared to other varieties at a given level of TW (Figure 2).

### 3.3. TW, TKW, KSD in Relation to Durum Milling Quality

Figure 4 illustrates the relationships between semolina and total milling yields to TW (a, b), TKW (c, d) and percentage of kernel fraction passing through no.6 slotted sieve (e, f). There were highly significant relationships (Figure 4a,b) between TW and milling yields (*R*^2^ = 0.75, *p* < 0.001) when results of all four varieties were combined. However, Transcend exhibited considerably lower semolina and total milling yields than other varieties across a wide range of TW examined (Table 2). At a given level of TW, Transcend had significantly lower TKW (Figure 2) and higher proportions of small kernels (Figure 3) than the other varieties. Since small kernels are detrimental to durum milling performance (Table 1), the higher proportion of small kernels in Transcend could be responsible for its inferior milling quality, such that affecting the overall relationship between TW and milling yields (Figure 4a,b).

To account for the impact of difference in KSD among varieties on durum wheat milling performance, the relationships between TKW (as a general indicator of kernel size) and milling yields were plotted in Figure 4c,d. Highly significant linear relationships were shown for semolina and total milling yields to TKW (R^2^ = 0.92, *p* < 0.001) regardless of durum variety selected. Samples with same TKW exhibited similar milling potential regardless of variety and TW. By account for the difference in KSD, TKW is a clearly a better predictor for durum wheat milling quality than TW for the selected samples in set II.

Nonetheless, different results have been reported for the relationships between kernel weight, kernel size and milling quality of durum wheat. Dexter et al. (1987) did not find kernel weight and kernel size alone or in combination with TW were better than TW in predicting durum milling quality due to the strong interdependence of TW and kernel size as suggested by the authors [3]. This study revealed that the interdependence between TW and TKW can be influenced by genotype (Figure 2) and is strongly related to KSD of the varieties (Figure 3). In addition, the coefficient of determination between wheat physical properties and semolina yield reported by Dexter et al. (1987) was much lower (*r*^2^ < 0.56) than the relationship in current study (*r*^2^ < 0.94), suggesting factors (e.g., genotype, environment and their interaction) other than kernel physical conditions contributed to semolina yield difference in the sample sets used by Dexter et al. (1987) [3].

### 3.4. The Potential of Kernel Size as a Grading Factor for Durum Wheat

Table 3 summarizes the inter-relationships between wheat protein content, TW, TKW, KSD and milling performance of selected durum samples in sample set II. In addition to the strong associations between TKW and milling yields (*r* > 0.96), highly significant correlations (*r* > 0.95) were found between milling yields and individual or combined kernel size fractions.

Overall, based on the detrimental impact of small kernels (<no.6) on durum milling performance (Table 1) and its strong negative relationship (*R*^2^ = 0.91, *p* < 0.001) with semolina and total milling yields (Table 3 and Figure 4e,f), the proportion of kernels passing through no.6 sieve appears to be promising as an objective grading factor for rapid prediction milling quality of durum wheat. Subsequently, threshold limits for the proportion of small kernels (passing no.6 sieve) can be established for grading CWAD. To protect milling quality of top grades of CWAD, the following threshold limits for proportion of small kernels were proposed: for no.1 and 2 CWAD: <30%; for no.3 CWAD: <40%; and for no.4 CWAD: <50%.

If these proposed tolerances were applied to samples in set II, Transcend 6, for example, would be downgraded to No.5 CWAD from No.3 due to presence of very high proportion of smaller kernels (58%) and inferior milling quality (Table 2), although it met the current TW requirement (77 kg/hL) for No.3 CWAD. On the contrary, Verona 4 and 5 would be upgraded to higher grades due to their high milling performance and relatively low percentage of smaller kernels despite their TW values did not meet current No.1 and No.2 CWAD requirements.

### 3.5. Yellow Pigment and Color Characteristics of Semolina and Pasta in Relation to Durum Kernel Size

Not only durum milling quality was affected by kernel size, significant effect of kernel size was shown for semolina and pasta color. Table 4 and Table 5 summarized the color characteristics of semolina and pasta made from sample sets I and II. To minimize the impact of flour on semolina color determination, measurement of color was conducted for semolina at both constant extraction rate of 70% and granules with removal of fine flour (<180 µm). Semolina prepared from small kernels exhibited higher ash content (Table 1 and Table 5) and duller color with low brightness (Table 4 and Table 5). The higher semolina ash of durum samples with low TW accounted for an additional 2.3% decrease in semolina yield per one kg/hL decrease in TW when semolina yield was calculated at a constant ash basis (Figure 1b). As ash content is often specified in semolina trading, durum wheat of larger kernels can achieve a greater extraction rate at constant ash specification, while it might be necessary to reduce semolina yield for durum of smaller kernel size in order to meet the ash specification.

In general, pasta prepared from samples of smaller kernels and low TW was significantly duller (lower L*) and showed greater redness (higher a*), especially for kernels passing through no.6 slotted sieve (Table 1) or TKW <35 g (Table 5). As pasta brownness/redness elevates with the increase of protein content [25], the significant increase in wheat protein as a result of low TW and TKW (Table 1 and Table 2) could be responsible for the variation in pasta redness.

Interestingly, TYP content increased significantly with the decrease of TW or TKW for all selected durum varieties (Table 5). This trend was also clearly demonstrated in Table 4 with the smallest kernels exhibited about 50% more TYP than the largest kernels (12.6 ppm vs. 8.1 ppm). However, the increase in TYP with the decrease of kernel size did not lead to a continuous increase in pasta yellowness (b* values, Table 4 and Table 5). Pasta yellowness improved with the increase of TYP up to a level, after which pasta yellowness reduced significantly even with higher TYP content. Analysis of grounded spaghetti showed that TYP of semolina and spaghetti was highly correlated (*R*^2^ = 0.92, *p* < 0.001), with an average TYP loss of 18% from semolina to dried pasta (Table 4 and Table 5). The decrease of pasta yellowness at higher TYP level suggests other underlying factors in affecting pasta yellowness. The relationships between semolina TYP, protein content and pasta yellowness showed that when wheat protein content is below 15%, the increase of TYP led to continuous improvement in pasta b*. However, further increase in wheat protein associated with small kernels resulted in a significant elevation in pasta redness (a* > 5.0) likely due to Maillard reaction [26,27]. The increase in pasta redness could adversely affect yellowness, leading to lower pasta b* values despite much higher TYP in small kernels. Thus, in order to improve pasta yellowness and maintain sufficient protein level for cooking quality, it is critical to understand the combined effects of protein content and TYP on pasta yellowness, particularly when high temperature drying is used in the processing.

### 3.6. Cooking Characteristics of Pasta Made from Durum with Different Kernel Sizes

Cooking quality of pasta is one of the most important factors in determining the end-use quality of durum wheat [28,29,30,31]. For the selected samples sets (I and II), semolina protein content exhibited highly significant relationship (*R*^2^ = 0.94, *p* < 0.001) to pasta firmness as measured by peak force across a wide range of durum samples evaluated (Figure 5). In other words, small kernels with high protein content possessed superior pasta cooking quality due to high firmness. Addition of gluten index (53 to 93%) as a second independent variable to semolina protein content did not improve the overall relationship to pasta firmness (*R*^2^ = 0.94, *p* < 0.001). There was no relationship between gluten index and pasta firmness (*R*^2^ = 0.01, *p* > 0.05). Dexter et al. (1987) stated that the lone beneficial effect of lower TW was increased protein content and improved cooked spaghetti firmness [3].

## 4. Conclusions

This study systematically evaluated the inter-relationships between TW, TKW, KSD and their effects on durum wheat milling, semolina composition and pasta processing quality. Except for higher protein content, smaller kernels exhibited lower semolina and total milling yields with higher semolina ash content. Detrimental impact on durum milling quality was shown for kernels passing through no.6 slotted sieve and particularly when calculated at constant ash basis. The relationship between TW and durum milling quality appeared to be affected by durum genotypes. At similar level of TW, variety Transcend showed consistently lower milling yields compared with other selected varieties due to its higher proportion of smaller kernels. By accounting for the difference in KSD, TKW or kernels passing no.6 slotted sieve showed great potential as an objective grading factor to better reflect milling quality of amber durum wheat. Work is in progress to verify the relationships between kernel size and milling quality by using multiple years of cargo samples. Threshold limits for small kernels (<no.6) have been proposed to Western Standard Committee as a new grading factor for CWAD. The much higher yellow pigment content in semolina milled from small kernels did not result a direct increase in yellowness in both semolina and pasta inferring the potential combined effect of protein content and yellow pigment on semolina and pasta color. Spaghetti prepared from samples with a high proportion of small kernels was firmer in texture but significantly duller and redder in appearance.

## Figures and Tables

**Figure 1 foods-09-01308-f001:**
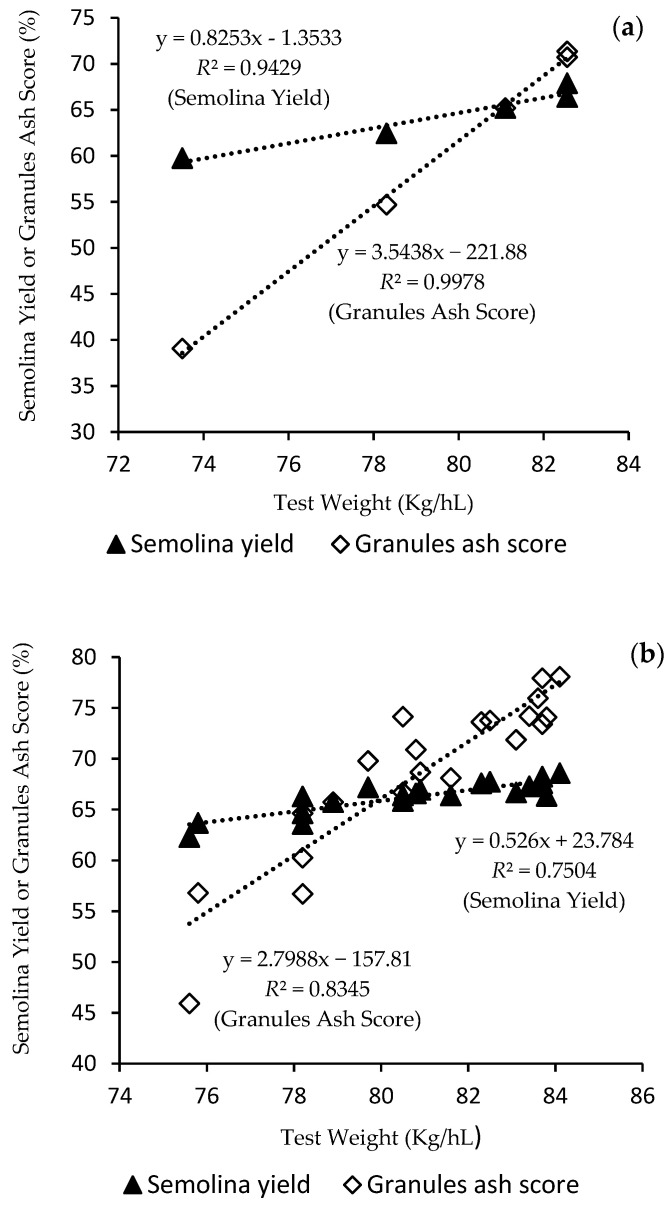
Impact of test weight on semolina yield and granules ash score for sample set I (**a**) and set II (**b**).

**Figure 2 foods-09-01308-f002:**
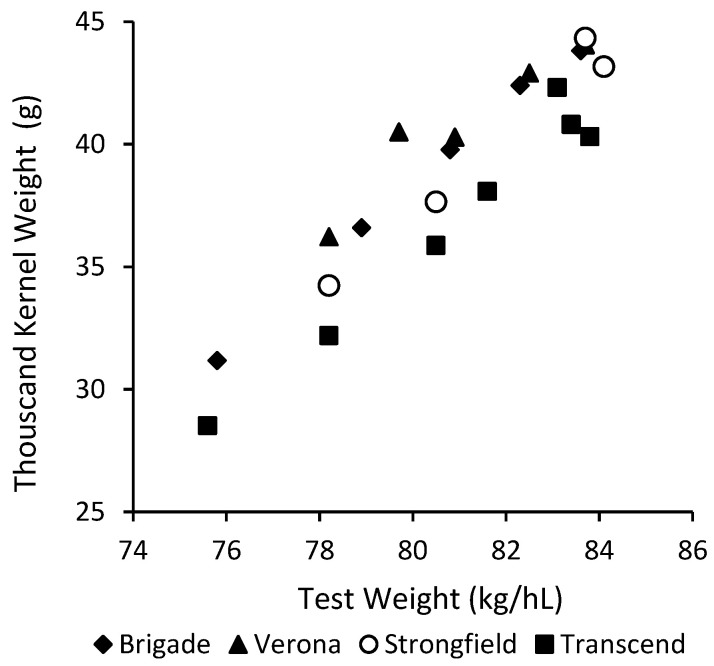
Inter-relationship between test weight and thousand kernel weight of varietal composites in sample set II.

**Figure 3 foods-09-01308-f003:**
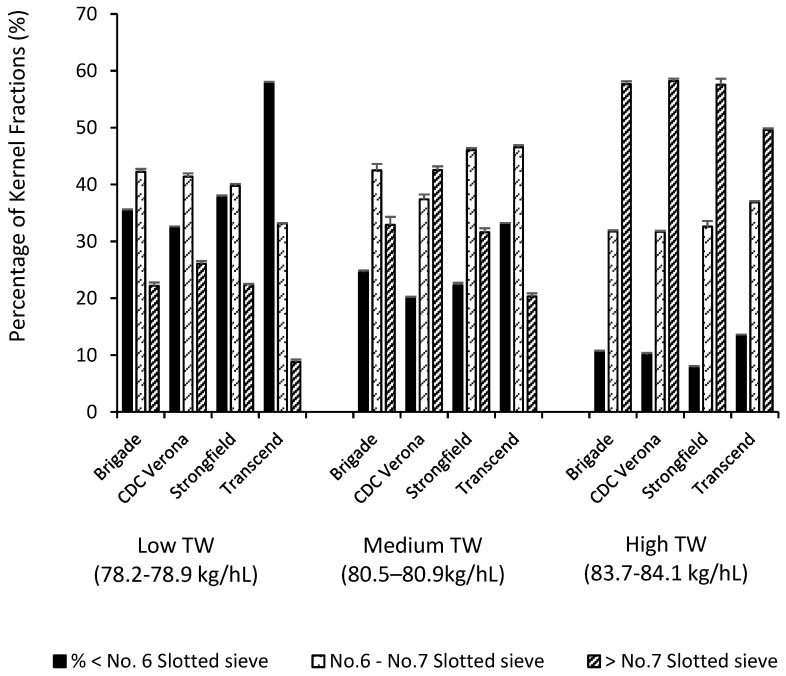
Impact of genotype and test weight on kernel size distribution of durum wheat.

**Figure 4 foods-09-01308-f004:**
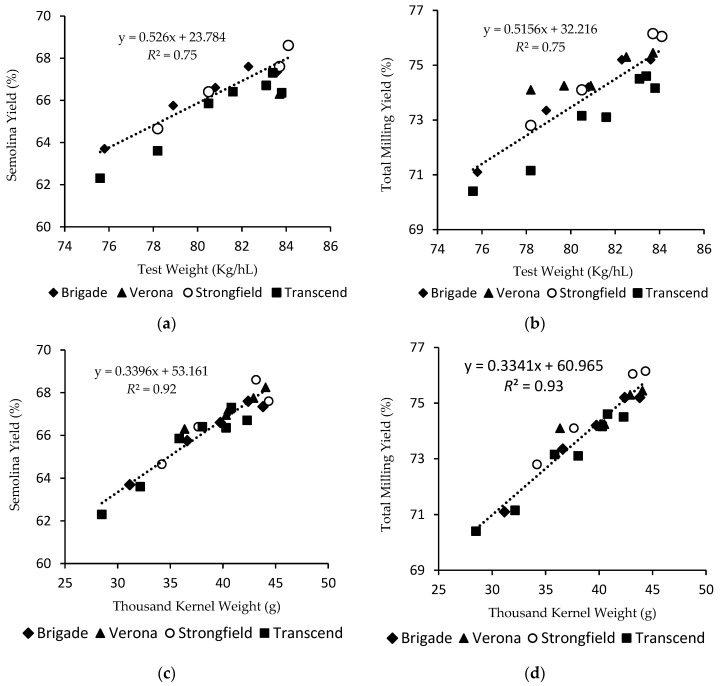
Relationships between semolina and total milling yields to test weight (**a**,**b**), thousand kernel weight (**c**,**d**) and proportion of kernels passing through slotted sieve no.6 (**e**,**f**).

**Figure 5 foods-09-01308-f005:**
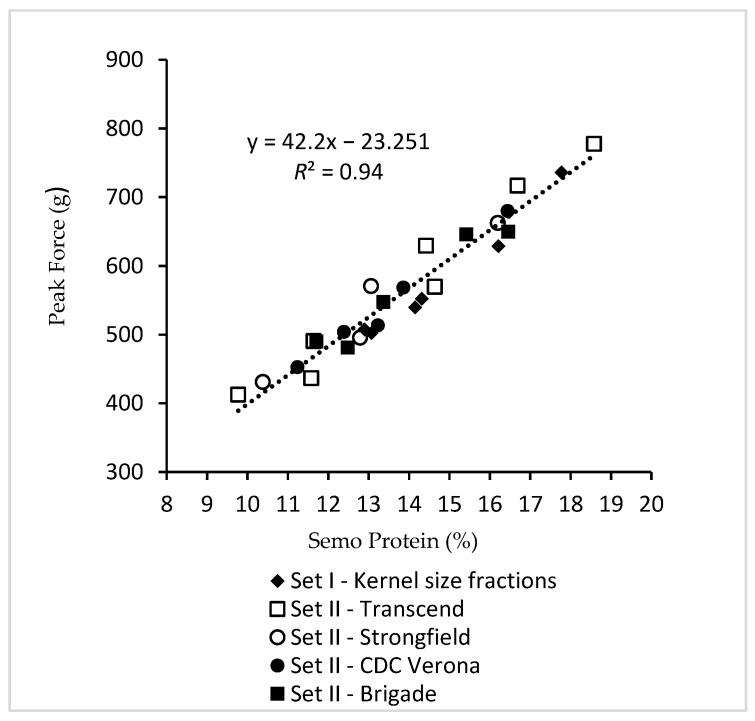
Relationships between semolina protein and spaghetti firmness as measured by peak force for samples in set I and set II.

**Table 1 foods-09-01308-t001:** Physical and milling properties of durum wheat samples fractionated based on kernel size (set I).

Sample Description	Grade	Wheat Protein (%)	TW (kg/hL)	TKW (g)	Wheat Ash (%)	Semolina Ash (%)	Semolina Yield (%)	Total Milling Yield (%)	Granule Ash Score (%)	Semolina Size Distribution (%)		
										<150 µm	150 < 250 µm	>250 µm
Unsorted Sample	1	15.1c	80.9b	37.7c	1.48d	0.70c	65.8bc	74.1b	65.8b	6.9d	25.4b	67.7b
>no.8 ^a^	2 ^b^	14.1d	82.6a	61.9a	1.52c	0.67d	67.9a	76.5a	71.4a	4.7e	25.4b	70.0a
>no.7	1	14.0d	82.6a	47.4b	1.46d	0.66d	66.4b	74.2b	70.7a	6.5d	25.6b	67.9b
>no.6	1	15.2c	81.1b	35.8d	1.49d	0.71c	65.2c	73.1c	65.2b	8.0c	25.4b	66.6c
>no.5	3 ^c^	17.1b	78.3c	27.8e	1.61b	0.80b	62.5d	70.2d	54.7c	11.2b	25.7b	63.1d
<no.5	5 ^d^	18.6a	73.5d	19.3f	1.86a	0.95a	59.8e	67.5e	39.1d	13.0a	26.6a	60.4e
*F* Value		2226.6 ****	4797.1 ****	12,482.5 ****	480.0 ****	525.7 ****	302.2 ****	477.1 ****	5232.7 ****	1749.9 ****	34.4 ***	701.7 ****

a–e, Mean values followed by same letter in the same column are not significantly different (*p* ≥ 0.05); ***, ****, indicate *F* value is significant at *p* < 0.001, 0.0001, respectively; TW, test weight; TKW, thousand kernel weight; ^a^ number of slotted sieves, aperture of no.5, 6, 7 and 8 slotted sieve equals to 1.98, 2.38, 2.78 and 3.18 mm, respectively; ^b^, Sample was downgraded due to light frost damage; ^c^, Sample was downgraded due to low test weight; ^d^, Sample was downgraded due to high level of shrunken kernels (% of kernels passed through 4^1/2^ slotted sieve)

**Table 2 foods-09-01308-t002:** Physical and milling properties of varietal composites prepared from 2017 harvest samples (set II).

Sample Description	Grade	Wheat Protein (%)	TW (Kg/hL)	TKW (g)	Kernel Size Distribution (%)						Semolina Yield (%)	Total Milling Yield (%)	Granules Ash Score (%)
					>no.8	>no.7	>no.6	>no.5	<no.5	<no.6			
Brigade 1	1	12.7m	83.6bc	43.8a	9.0b	48.7ab	31.8k	9.0m	1.5kl	10.6mn	67.4bcde	75.2abc	76.0b
Brigade 2	1	13.5k	82.3f	42.4abc	6.9cd	41.2e	36.6ghi	13.3j	2.0k	15.3k	67.6bcd	75.2abc	73.6c
Brigade 3	1	14.4i	80.8h	39.8de	3.8fgh	29.1g	42.5b	21.1g	3.5fg	24.6h	66.6efghi	74.2de	70.9e
Brigade 4	3 ^a^	16.4e	78.9k	36.6fg	3.5fgh	18.7i	42.3bc	29.7d	5.8d	35.5e	65.8i	73.4efg	65.8i
Brigade 5	5 ^b^	17.4c	75.8m	31.2j	0.9ij	11.0j	27.9l	41.7b	18.5b	60.2b	63.7k	71.1h	56.8l
Verona 1	1	12.1o	83.7b	44.1a	10.9a	47.3abc	31.7k	8.6m	1.5kl	10.1m	68.3ab	75.5ab	73.4c
Verona 2	1	13.1l	82.5e	42.9ab	6.6de	44.6cd	34.8hij	11.6k	2.4ij	14.0l	67.8abc	75.3ab	73.8c
Verona 3	1	14.0j	80.9h	40.3cd	4.8ef	37.7f	37.4efgh	17.3i	2.7hi	20.1j	67.0cdefg	74.3cde	68.7g
Verona 4	2 ^a^	14.8h	79.7j	40.5cd	4.3fg	36.3f	39.6cdef	16.7i	3.1gh	19.8j	67.2cdefg	74.3cde	69.8f
Verona 5	3 ^a^	17.1d	78.2l	36.2fgh	1.9ghi	24.1h	41.4bc	27.4e	5.1e	32.5f	66.3ghi	74.1def	60.3k
Strongfield 1	1	11.3p	84.1a	43.2a	8.7bc	48.9ab	32.6jk	8.3m	1.6kl	9.9n	68.6a	76.1a	78.1a
Strongfield 2	1	13.9j	83.7b	44.3a	11.5a	49.0a	31.6k	6.4n	1.4l	7.8o	67.6bcd	76.2a	77.9a
Strongfield 3	1	13.9j	80.5i	37.6efg	2.5ghi	29.1g	46.1a	19.8h	2.5hij	22.3i	66.4fghi	74.1de	74.2c
Strongfield 4	3 ^a^	17.1d	78.2l	34.2hi	2.3hij	20.2i	39.8bcde	32.2c	5.7de	37.8d	64.7j	72.8g	64.7j
Transcend 1	1	10.6q	83.8b	40.3cd	7.4bcd	42.2de	36.9fghi	12.0k	1.6kl	13.5l	66.4fghi	74.2de	74.1c
Transcend 2	1	12.5n	83.4c	40.8bcd	6.9cd	41.3e	38.1defg	12.3k	1.5kl	13.8l	67.3cdef	73.1g	74.2c
Transcend 3	1	15.4g	83.1d	42.3abc	8.6bcd	45.8bc	34.3ijk	10.3l	1.1l	11.3m	66.7defgh	74.6bcd	71.9d
Transcend 4	1	12.6mn	81.6g	38.1ef	3.2fgh	29.0g	40.8bcd	23.3f	3.7f	27.1g	66.4fghi	73.2fg	68.1g
Transcend 5	1	15.6f	80.5i	35.9gh	1.0ij	19.3i	46.6a	29.6d	3.5fg	33.1f	65.9hi	74.5bcd	66.7h
Transcend 6	3 ^a^	17.6b	78.2l	32.2ij	0.6j	8.3jk	33. jk1	46.4a	11.6c	58.0c	63.6k	71.0h	56.7l
Transcend 7	5 ^b^	19.3a	75.6m	28.5k	0.5j	6.7k	26.3l	47.3a	19.2a	66.5a	62.3l	70.4h	45.9m

a–o, For each variety, mean values followed by same letter in the same column are not significantly different (*p* ≥ 0.05); TW, test weight; TKW, thousand kernel weight; ^a^, Samples were downgraded due to low test weight; ^b^, Samples downgraded due to increased amount of shrunken kenrels (Brigade 5: 4.6%; Transcend 7: 5.1%).

**Table 3 foods-09-01308-t003:** Inter-relationships between durum wheat physical properties and milling quality (set II).

	TW	TKW	Semolina Yield	Total Milling Yield
Wheat Properties				
Protein	−0.96 ***	−0.94 ***	−0.93 ***	−0.91 ***
TW	-	0.92 ***	0.87 ***	0.87 ***
TKW	0.92 ***	-	0.97 ***	0.96 ***
Kernel Fractions				
>no.8	0.88 ***	0.90 ***	0.79 ***	0.85 ***
>no.7	0.93 ***	0.97 ***	0.91 ***	0.93 ***
>no.6	0.03 ^ns^	0.01 ^ns^	0.19 ^ns^	0.13 ^ns^
>no.5	−0.93 ***	−0.98 ***	−0.94 ***	−0.96 ***
<no.5	−0.86 ***	−0.88 ***	−0.89 ***	−0.88 ***
Combined Kernel Fractions				
>no.7	0.93 ***	0.97 ***	0.90 ***	0.93 ***
>no.6, <no.8	0.90 ***	0.95 ***	0.95 ***	0.94 ***
>no.5, <no.7	−0.86 ***	−0.90 ***	−0.81 ***	−0.85 ***
<no.6	−0.93 ***	−0.97 ***	−0.95 ***	−0.96 ***

***, indicate correlation coefficients between parameters is significant at *p* < 0.001, respectively; ns, not significant (*p* ≥ 0.05); TW, test weight; TKW, thousand kernel weight.

**Table 4 foods-09-01308-t004:** Semolina and spaghetti color characteristics of durum wheat samples prepared based on kernel size.

Sample Description	Wheat Protein	Semolina Protein (%)	Semolina (70% Extraction)				Semolina (>180 um)			Spaghetti			
			TYP (ppm)	L*	a*	b*	L*	a*	b*	TYP (ppm)	L*	a*	b*
All kernels	15.1c	14.2c	10.0d	83.6b	−2.6cd	32.3bc	82.9b	−2.6cd	34.5b	8.1d	72.2b	5.2c	64.2b
>no.8 ^∆^	14.1d	12.9d	8.1f	84.1a	−2.7de	30.7d	83.7a	−2.7cd	32.2c	6.6f	73.6a	3.9d	62.1c
>no.7	14.0d	13.1d	9.0e	84.1a	−2.8e	31.9c	83.4ab	−2.7d	34.1b	7.6e	73.6a	4.0d	64.6ab
>no.6	15.2c	14.3c	10.3c	83.7b	−2.6c	33.1a	82.9b	−2.5c	35.2ab	8.7c	72.2b	5.2c	65.3a
>no.5	17.1b	16.2b	11.5b	83.3b	−2.4b	32.8ab	82.0c	−2.2b	36.0a	9.9b	70.2c	7.0b	64.6ab
<no.5	18.6a	17.8a	12.6a	82.6c	−2.0a	32.1bc	81.0d	−1.7a	35.6a	10.8a	66.7d	9.9a	61.7c
*F* value	2226.6 ****	1736.2 ****	6423.4 ****	63.3 ****	139.8 ****	58.0 ****	175.9 ****	155.5 ****	58.4 ****	4274.7 ****	1340.1 ****	1473.3 ****	98.3 ****

a–f, mean values followed by same letter in the same column are not significantly different (*p* ≥ 0.05); ****, indicate *F* value is significant at *p* < 0.0001, respectively; TYP, total yellow pigment; L*, brightness; a*, redness; b*, yellowness; ^∆^ number of slotted sieves, aperture of no.5, 6, 7 and 8 slotted sieve equals to 1.98, 2.38, 2.78 and 3.18 mm, respectively.

**Table 5 foods-09-01308-t005:** Semolina and spaghetti color characteristics of varietal composites prepared from 2017 harvest samples.

Sample Description	Wheat Protein (%)	Semolina Protein (%)	Wheat Ash (%)	Semolina Ash (%)	Semolina (70% Extraction)				Semolina (>180 µm)			Spaghetti			
					TYP (ppm)	L*	a*	b*	L*	a*	b*	TYP (ppm)	L*	a*	b*
Brigade 1	12.7m	11.7m	1.31kl	0.61hi	8.2h	84.3bc	−2.8hi	30.6gh	83.6b	−2.9j	33.0def	6.5i	74.3	3.1	61.6
Brigade 2	13.5k	12.5l	1.36j	0.64fgh	8.6e	83.9defg	−2.7fg	31.4bcd	83.4bc	−2.7ij	33.4cde	7.1f	73.4	3.6	62.6
Brigade 3	14.4i	13.4i	1.38i	0.66f	9.5b	83.8defgh	−2.7fg	32.3a	83.0cde	−2.6ghi	34.4ab	7.5d	72.7	4.8	63.6
Brigade 4	16.4e	15.4e	1.41fg	0.71cd	10.1a	83.4hijk	−2.3cd	32.0ab	82.5efg	−2.2cd	34.7a	8.2ab	71.5	6.0	63.5
Brigade 5	17.4c	16.5c	1.51d	0.79b	10.2a	83.5fghij	−2.4de	31.6bcd	82.2gh	−2.2bc	34.8a	8.3a	70.3	7.2	62.1
Verona 1	12.1o	11.2n	1.38hi	0.65fg	8.1hi	84.0cde	−2.7fg	31.3cde	83.4bc	−2.7hi	32.7f	6.6hi	74.0	3.4	61.6
Verona 2	13.1l	12.4l	1.34j	0.64fgh	8.6ef	83.9def	−2.7fg	31.3cdef	83.2bcd	−2.6fghi	33.5cde	7.1f	73.2	3.6	62.9
Verona 3	14.0j	13.2ij	1.43ef	0.69de	8.9d	83.4hijk	−2.5e	31.6abc	82.9cde	−2.4efg	33.5cd	7.2f	72.6	4.9	62.0
Verona 4	14.8h	13.9h	1.40gh	0.68e	8.8de	83.7efghi	−2.5e	31.0defg	82.9cde	−2.4de	33.1def	7.2f	72.3	5.2	62.1
Verona 5	17.1d	16.4c	1.55c	0.78b	9.3c	83.3jk	−2.2bc	30.5ghi	82.2gh	−2.0b	33.5cde	7.3ef	70.8	6.5	61.1
Strongfield 1	11.3p	10.4o	1.30l	0.60ij	7.7j	84.3bc	−2.9hi	30.3hi	83.6b	−2.7ij	31.9h	5.9k	74.2	3.3	59.2
Strongfield 2	13.9j	12.8k	1.32kl	0.59j	7.7j	84.0bcd	−2.7gh	29.9i	83.4bc	−2.6ghi	31.9h	6.1j	74.3	3.0	60.3
Strongfield 3	13.9j	13.1j	1.31kl	0.62gh	8.4fg	83.8defg	−2.5ef	30.5ghi	83.2bcd	−2.5efgh	32.7fg	6.8g	72.4	4.4	61.9
Strongfield 4	17.1d	16.2d	1.51d	0.71c	8.8de	83.4ijk	−2.2b	30.3hi	82.5efg	−2.1bc	32.8f	6.9g	71.2	6.0	60.7
Transcend 1	10.6q	9.8p	1.31kl	0.62hi	8.1hi	84.7a	−2.9i	30.7fgh	84.1a	−2.8j	32.7fg	6.8h	74.3	3.5	61.7
Transcend 2	12.5n	11.6m	1.33k	0.63fgh	8.3gh	84.4ab	−2.8ghi	30.6gh	83.5b	−2.7hi	32.9ef	7.1f	73.7	3.7	62.9
Transcend 3	15.4g	14.4g	1.41fg	0.65f	8.0i	83.9de	−2.4de	30.3ghi	83.3bcd	−2.4def	32.2gh	6.8gh	72.9	4.3	61.7
Transcend 4	12.6mn	11.6m	1.44e	0.69de	8.8de	84.3bc	−2.7gh	31.4bcd	83.4bc	−2.6ghi	33.5cde	7.4de	72.9	4.7	62.7
Transcend 5	15.6f	14.6f	1.41fg	0.70cde	9.5bc	83.8defg	−2.5e	31.4bcd	82.8def	−2.3de	33.9bc	7.8c	72.0	5.2	63.4
Transcend 6	17.6b	16.7b	1.58b	0.79b	9.5b	83.5ghijk	−2.1b	30.8efgh	82.3fg	−2.0b	33.8c	8.1b	70.7	6.5	62.6
Transcend 7	19.3a	18.6a	1.79a	0.90a	10.0a	83.2k	−1.9a	30.6fgh	81.8h	−1.7a	33.8c	8.2a	69.0	8.3	61.2

a–n, mean values followed by same letter in the same column are not significantly different (*p* ≥ 0.05). TW, test weight; TKW, thousand kernel weight; TYP, total yellow pigment.

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
