# Peer review of "Inter-Relationships between Test Weight, Thousand Kernel Weight, Kernel Size Distribution and Their Effects on Durum Wheat Milling, Semolina Composition and Pasta Processing Quality"

_foods, 2020, doi:10.3390/foods9091308_

Round 1
Reviewer 1 Report
Evaluation of the article: Inter-relationships between Test Weight, Thousand Kernel Weight, Kernel Size Distribution and Their Effects on Durum Wheat Milling, Semolina Composition and Pasta Processing Quality, by Kun Wang and Bin Xiao Fu
2nd round.
The article has been improved. Nevertheless some changes need to be done:
Line 152 change:
Deceased to Decreased
Line 249:
Table 3 summarized … to Table 3 summarizes…
Correctly format Ref. 4
Author Response
Dear Reviewer 1,
Your comments are greatly appreciated. All your comments have been carefully considered and revisions made accordingly.
Reviewer 1
The article has been improved. Nevertheless some changes need to be done:
Line 152 change:
Deceased to Decreased (Changed)
Line 249:
Table 3 summarized … to Table 3 summarizes… (Changed)
Correctly format Ref. 4 (Reformatted)
Reviewer 2 Report
Second review of revised Foods-886118, Wang and Fu
Re. Sample Set 1. It is still not clear to me what/where were the replications.
I understand that there was one composite, and that it was fractionated into 5 sub-fractions.
Running multiple (duplicate?) measurements on each of these sub-fractions and the unsorted sample does not constitute an appropriate design for ANOVA in terms of the objective. Yes, the fractions are generally different from one another. Were multiple lots of each sub-fraction milled separately?
As it stands, the results are not wrong by any means, but are limited to this one composite and how it fractionated into sub-fractions. This provides limited ability to extrapolate the results to any other possible cargo composite. In my opinion, the results are “valid” but should be presented with the recognition of the foregoing limitations. Duplicate measurements do not really address true error variance, only the variation due to sampling and analytical error. Probably some sort of response “curve” with standard deviation error bars would be more appropriate than ANOVA.
Similarly, it is still not clear to me the design for Set II. Among the 21 composites, how many were comprised of Transcend? Or Brigade? Or Strongfield? Or CDC Verona? Was it five each and one with six? Again, I have no reason to question the results of the analytical measurements, but I disagree with the statistical analysis and presentation/interpretation of results. Once again, where is the replication? You have created multiple samples for each cultivar, but those are confounded with the selection criteria (test weight and protein).
I am suggesting that the authors re-work their research with the foregoing comments taken in earnest, rather than simply defend what they have already presented.
Author Response
Dear Reviewer 2,
Your comments are greatly appreciated. We revised the manuscript according to your comments and further explanation provided on the sample sets.
Re. Sample Set 1. It is still not clear to me what/where were the replications.
I understand that there was one composite, and that it was fractionated into 5 sub-fractions.
Running multiple (duplicate?) measurements on each of these sub-fractions and the unsorted sample does not constitute an appropriate design for ANOVA in terms of the objective. Yes, the fractions are generally different from one another. Were multiple lots of each sub-fraction milled separately?
Yes, please see Line 92: All durum samples were milled into semolina in duplicate of 1.5 kg lots separately using…
As it stands, the results are not wrong by any means, but are limited to this one composite and how it fractionated into sub-fractions. This provides limited ability to extrapolate the results to any other possible cargo composite. In my opinion, the results are “valid” but should be presented with the recognition of the foregoing limitations. Duplicate measurements do not really address true error variance, only the variation due to sampling and analytical error. Probably some sort of response “curve” with standard deviation error bars would be more appropriate than ANOVA.
Added in Lines 72-73 (material and methods): representing over half million metric tons of durum wheat in a shipping period of eight months.
Added in Lines 338-340 (conclusions): Work is in progress to verify the relationships between kernel size and milling quality by using multiple years of cargo samples.
Similarly, it is still not clear to me the design for Set II. Among the 21 composites, how many were comprised of Transcend? Or Brigade? Or Strongfield? Or CDC Verona? Was it five each and one with six? Again, I have no reason to question the results of the analytical measurements, but I disagree with the statistical analysis and presentation/interpretation of results. Once again, where is the replication? You have created multiple samples for each cultivar, but those are confounded with the selection criteria (test weight and protein).
Table 2 shows that the number of composites for each variety.
Line 80-81: Each composite was prepared from a minimum of 15 samples.
Sample set II was used to investigate the relationship between test weight, thousand kernel weight, kernel size distribution and quality parameters of four major Canadian durum varieties. It was generated by screening over 1,200 samples submitted by the durum producers in Canada. It is not practical to grow in test plots for such a set with so big variations in test weight, thousand kernel weight, and protein content.
This manuscript is a resubmission of an earlier submission. The following is a list of the peer review reports and author responses from that submission.
Round 1
Reviewer 1 Report
Evaluation of the article: Inter-relationships between Test Weight, Thousand Kernel Weight, Kernel Size Distribution and Their Effects on Durum Wheat Milling, Semolina Composition and Pasta Processing Quality, by Kun Wang and Bin Xiao Fu
General comment:
This work describes the properties of wheat varieties produced in Canada in 2017 and 2018. Most of them met the test weight (TW) requirements for the top grades of Canada Western Amber Durum (CWAD), while some samples of top grades were inferior in milling quality. Samples having lower milling yields had greater proportion of smaller kernels.
The article is well written and the experiments logically planned and described.
The relationship of milling quality with weight and volume properties is investigated suggesting that Kernel Weight (KW) is a best indicator of milling quality than Test Weight (TW). The interdependence between TW and TKW can be influenced by genotype and is strongly related to kernel size distribution (KSD) of the varieties. The authors propose to establish threshold limits for the proportion of small kernels (passing no.6 sieve) for grading CWAD.
It could be interesting if the authors could, in this or future experiments, check the shape of kernels. In European samples it has been shown that the images of kernels of Triticum durum adjust well to an ellipse of Aspect Ratio= 2.4, and it may be interesting to see if this proportion changes in varieties with smaller kernels, or in the small kernels of some of the varieties tested. Please see: https://www.mdpi.com/2073-4395/9/7/399/htm
Comments by sections:
Materials and methods
Tables 1 and 2: please explain the terms Grade and Grade (Export).
Results and Discussion
3.1. Impact of kernel size on durum wheat milling quality
Line 182, add space in:
Figure1a,
Convert to: Figure 1a,
Lines 184-187, please try to write more clear and concisely:
the increase of semolina ash content associated with smaller kernel size led to about 3.5% decrease in semolina yield with each one kg/hL decrease in TW, significantly higher than 0.8% when semolina yield was calculated without considering semolina ash differences.
In the middle of Figure 1a, please correct:
(Granules ash socre)
To:
(Granules ash score)
3.2 Effect of genotype on the relationship between TW, TKW and KSD
Please arrange Figure 3 so all the legends at the foot can be read completely:
Low TW
(78.2-78.9 kg/hL)
Medium TW
(80.5–80.9kg/hL)
High TW
(83.7-84.1 kg/hL)
And correct:
% < No. 6 Slotted sieve
No.6 - No.7 Slotted sieve
> No.7 Slotted sieve
3.3 TW, TKW, KSD in relation to durum milling quality
Lanes 220-223: Please make clear the following:
Although there were highly significant relationships (Figure 4a, 4b) between TW and milling yields for individual durum variety (R2 = 0.86, p < 0.05), such relationship was weaker (R2 = 0.75, p < 0.001) when all varieties were included.
To what variety does the above sentence refer? What variety gives R2 = 0.86?
In addition, check the results corresponding to this sentence: such relationship was weaker (R2 = 0.75, p < 0.001) when all varieties were included. Because p < 0.001 does not mean weaker than p < 0.05.
3.4 The potential of kernel size as a grading factor for durum wheat
3.5 Yellow pigment and color characteristics of semolina and pasta in relation to durum kernel size b
Lane 296, please change:
b* values (Tables 4 and 5).
to:
yellowness (b* values; Tables 4 and 5).
References:
Please check the correct format in all references. For example correct:
Baker, D., Fifield, C.C., & Hartsing, T. F. Factors related to the flour-yielding capacity of wheat. Northwest Miller, 1965, 272, 16-18.
To:
Baker, D.; Fifield, C.C.; & Hartsing, T. F. Factors related to the flour-yielding capacity of wheat. Northwest Miller, 1965 272, 16-18.
Reviewer 2 Report
Review of Foods foods-886118, Wang and Fu
The paper presents interesting results related to durum wheat quality using a composite sample, and set of pure variety composites.
Unfortunately, the first sample (“Set I”) does not seem to be a “set”, but rather a single sample derived from compositing 14 commercial cargo samples. As such, it has limited value, and there is no basis for running ANOVA. The fact that analyses were run in duplicate does not constitute replication. Consequently, there is no basis for providing mean separation and F-values in Table 1. These are simply the values obtained for this one composite.
Set II consisted of 21 composites of four cultivars, each composite derived from a minimum of 15 sub-samples. Similarly, it is a bit of a stretch to consider any of these as ‘replicates’. I would challenge the authors to justify the basis of running ANOVA on these. It is not clear what the experimental design was that generated the mean separations in Table 2. This must be explained and then re-evaluated. The results in Table 3 strike me as a bit strange in that there are highly significant correlation coefficients for > no. 7, > no. 5, and < no. 5, but those for > no. 6 are nearly zero.
The English needs to be improved. There is in general, an excessive number of digits. Parts of figures are cut-off and not legible. In two instances the text refers to “previous studies” (line 224) and “a subsequent study” (line 271) with no reference provided.